# Outpatient care changes and associated mortality among Veterans with heart failure during the COVID-19 pandemic

**Shilpa Vijayakumar**[1], **Emily Corneau**[2], **Sebhat Erqou**[2,3], **Aravind Kokkirala**[3], **Wen-Chih Wu**[2,3]*

**1** Cardiovascular Imaging Program, Cardiovascular Division and Department of Radiology, Brigham and Women's Hospital and Harvard Medical School, Boston, Massachusetts, United States of America, **2** Center of Innovation, Providence VA Medical Center, Providence, Rhode Island, United States of America, **3** Department of Medicine, Providence VA Medical Center, Alpert Medical School of Brown University, Providence, Rhode Island, United States of America

* wen-chih.wu@va.gov

## Abstract

### Background

The mortality risk associated with loss of in-person outpatient visits or transition to virtual care in patients with heart failure (HF) during the COVID-19 pandemic is unknown.

### Objectives

Assess changes in outpatient HF care patterns and associated mortality.

### Methods

Retrospective analysis of HF patients using national Veterans-Health-Administration (VHA) data. Among 509,511 HF patients who received VHA care, we compared mean monthly days-with-an-outpatient-visit from 2/2018–1/2020 (pre-COVID) versus 2/2020–1/2021 (COVID) using T-tests. In a subset of 321,439 patients with ≥1 VHA cardiology or primary-care visit in 2019, we related the presence and type of outpatient visit with mortality using Cox-Regression estimated hazard-ratios (HRs).

### Results

Despite a 2–3-fold increase in video-only visits and use of telephone visits to maintain access, the overall days with outpatient visits decreased from a monthly-average of 81.4±6.1 in 2018–2019 and 81.0±5.6 in 2019–2020, to 57.8±11 days in 2020–2021 (P<0.01 for both), per 100 Veterans. When compared to patients with no-visits during the study period, the adjusted-mortality risk was lowest for patients with at least one in-person (HR 0.42, 95%CI: 0.41–0.44), followed by video-only (HR 0.52,

**Data availability statement:** The data underlying this study cannot be shared publicly because they contain sensitive patient information and are restricted under the Health Insurance Portability and Accountability Act (HIPAA). However, data are available upon request for researchers who meet the necessary ethical and legal requirements. Requests for data access can be made to the Providence VA Healthcare System Institutional Data Access/Ethics Committee by contacting Amy Mochel via email at amy.mochel@va.gov.

**Funding:** Department of Veterans Affairs, HSRD funding IRP 20-003. The funders had no role in study design, data collection and analysis, decision to publish, or preparation of the manuscript.

**Competing interests:** Dr. Wu and Emily Corneau have no conflicts to declare. Dr. Vijayakumar is supported by NHLBI T32 postdoctoral training grant T32HL094201and a research grant from the Amyloidosis Foundation. Dr. Erqou is supported by the Department of Veterans Affairs, VISN-1 Career Development Award. Dr. Kokkirala owns shares in TDOC and AMWL telehealth stocks. There are no relationships with industry. This does not alter our adherence to PLOS ONE policies on sharing data and materials.

**Abbreviations:** COVID-19, Coronavirus disease-2019; CI, Confidence interval; EF, Ejection fraction; HR, Hazard ratio; HF, Heart failure; ICD, International Classification of Diseases; VA, Veterans Affairs; VHA, Veterans Health Administration.

95%CI: 0.50–0.55) and then telephone-only (HR 0.57, 95%CI: 0.54–0.60) visits (p = 0.14 for trend). Results remained similar when the analysis was repeated (without including telephone visits) for pre-COVID (2/2018–1/2020) periods.

## Conclusions

Despite an increase in video and use of telephone visits during the COVID-19 pandemic, there was still a decrease in total outpatient visits for patients with HF. The presence and type of outpatient encounter was associated with the adjusted risk of mortality.

## Introduction

Coronavirus disease-2019 (COVID-19) has affected millions around the world, and patients with underlying heart failure (HF) are particularly vulnerable to its effects [1,2]. Beyond its direct impact, the COVID-19 pandemic has also resulted in a global transformation in the provision and delivery of healthcare, including temporal deferral of in-person outpatient visits and shifts to virtual video visits and telephone calls [3]. Despite the end of the COVID-19 public health emergency declaration, telehealth has now become a fixture of healthcare delivery in the United States, and the Centers for Medicare and Medicaid (CMS) have now considerably expanded the list of services which can be provided by telehealth. CMS has extended the time many of these services will be covered to the end of 2024 and has made coverage of some telehealth services permanent [4]. Prior studies have shown that the reduction in outpatient visits during the COVID-19 pandemic has led to delays in cancer care [5], but its effects in patients with chronic, severe cardiovascular diseases, such as HF, have not been reported. Patients with HF may be immediately and disproportionately affected by missed outpatient follow-up appointments. Patients with HF require close outpatient follow-up to assess changes in volume status, optimize evidence-based pharmacologic therapies and prevent worsening HF [6]. Challenges with virtual care may also be heightened with HF patients in whom lack of physical examination and technical difficulties with online communication may potentially result in inadequate treatment and poorer prognosis [3,7]. Thus, decreases in in-person clinical encounters, if any, may result in adverse outcomes including higher mortality in vulnerable HF patients. However, pandemic related changes to outpatient HF care and associated outcomes have not been fully explored. In particular, with the possibility of recurrence of similar epidemics/pandemics that could potentially disrupt healthcare system, there is an urgent unmet need to identify ways to best care for patients with severe, complex, chronic disease, such as HF, in the future. Additionally, as digital technologies and telemedicine continue to expand, the impact of virtual video or telephone visits on complex outpatient care remains unclear. The purpose of this study was to analyze changes in outpatient clinical care, including the transition to virtual video or telephone care, of Veterans with HF during the COVID-19 pandemic, and to assess the relationship of these changes with mortality.

## Methods

### Study design

We conducted a retrospective study using cross-linked Department of Veterans Affairs (VA)-electronic health record data of a nationwide cohort of Veterans diagnosed with HF prior to the COVID-19 pandemic onset. We determined the change in volume of in-person and virtual video or telephone visits of these patients before and during the COVID-19 pandemic and compared types of outpatient visits to understand the association of changes in outpatient visit to mortality.

This study was approved by the Institutional Review Board (IRB) of the Providence VA Medical Center, under the project name "Quality of Care and Outcomes in Heart Failure", originally on 11/12/2008, exempt from ongoing IRB review on 6/14/2019, with the most recent institutional research committee approval on 3/20/2024. Data was accessed for this research project on 5/3/2021, the authors had access to information that could identify individual participants after data collection to cross-link multiple health information and utilization data across different VA datasets.

### Study sample and cohort assembly

Data sources included VA enrollment files, inpatient and outpatient encounters, and COVID-19 Shared Data Resource [8]. Veterans with HF were identified based on an inpatient or outpatient visit with a HF diagnosis between February 1, 2016-January 31, 2020, with an international classification of disease, ninth revision (ICD-9), or tenth revision (ICD-10) codes for HF diagnosis (n = 509,511). For analytical purposes, we operationally defined February 1, 2018 to January 31, 2019, and February 1, 2019 to January 31, 2020 as *pre-pandemic baseline* periods and February 1, 2020 to January 31, 2021 as the COVID-19 pandemic period. February 1, 2020, was chosen as the start of the COVID-19 pandemic period in our study based on the first declaration of COVID-19 as a public health emergency in the U.S. issued on January 31, 2020, under Section 319 of the Public Health Service Act.

### Pre- and pandemic period outpatient healthcare utilization and all-cause mortality

Yearly sample sizes ranged from 323,617 in 2/2018–1/2019, 369,384 in 2/2019–1/2020 and 354,092 in 2/2020–1/2021, for a total of 509,511 unique patients analyzed. To contrast outpatient care between pre- and the pandemic periods, we calculated the number of days with any outpatient clinical encounter, either in-person, video-only or telephone-only, per 100 people alive at the beginning of each month, and we graphed monthly rates in the volume of these encounters among our cohort beginning with the *pre-pandemic baseline* periods (2/2018–1/2019 and 2/2019–1/2020), through one year into the COVID-19 pandemic (2/2020–1/2021).

Similarly, we calculated the number of deaths per 1000 people alive at the beginning of each month, among our cohort over time, and graphed monthly mortality rates beginning with the *pre-pandemic baseline* periods (2/2018–1/2019 and 2/2019–1/2020), through one year into the COVID-19 pandemic (2/2020–1/2021).

### In-person, video-only or telephone-only encounters during the pandemic and relationship with mortality

**Exposure.** To relate the presence and type of clinical encounter with mortality, we restricted our sample to patients diagnosed with HF and cared for at the Veterans Health Administration (VHA) between 2/2016–1/2020 (n = 509,511) who had at least 1 cardiology or primary care visit in 2019 (82,091 excluded) and were alive by 2/1/2020 (105,981 excluded) in the VHA yielding a final analytic sample of 321,439.

The main exposure period was defined as February 1, 2020 to September 30, 2020 (or until death). The end of the exposure period was chosen based on the VA decree issued by the Office of Inspector General (Report # 20-02794-218) to declare that the VHA will take action to ensure follow-up of cancelled outpatient appointments by the end of September 2020 [9].

The main exposure was the number of days with at least one clinical encounter (either in-person or video-only or telephone-only visit) that occurred during the exposure period, operationally named, as **"outpatient visit days"**, a continuous variable. We chose "days" as opposed to "encounters" as the unit of exposure because patients may have multiple clinical encounters and of different types within the same day. These included VA outpatient clinical encounters comprising a broad range of disciplines, such as cardiology, primary care, physical therapy, and social work (S1 Data). Additionally, we determined if the outpatient encounter was in-person, video-only/telehealth, or telephone-only (S1 Data). In cases where multiple clinical encounters of different types (e.g., in-person, video and telephone) occurred within the same day, we used the encounter with the highest intensity of contact (in-person > video-only > telephone-only) as the visit type for the day.

We stratified the patients based on whether they had any in-person (may contain days with video or telephone encounters), video-only (no in-person days but may contain telephone), telephone-only (no days with in-person or video), or no (virtual or in-person) outpatient encounter days during the exposure period and related the type of outpatient encounter to mortality beginning on February 1, 2020.

## Outcome

Patients were followed for all-cause mortality during the *outcome period* (February 1, 2020, to May 31, 2021). Dates of death were confirmed using the VA's Vital Status File as well as the Master Patient Index, [10] which contain mortality data from multiple VA and non-VA data sources with a validated 98.3% sensitivity and 97.6% agreement with the National Death Index [11].

## Historical cohorts

To assess if the relationship between visit types and mortality was also present in pre-COVID times, we repeated our analysis of visit-type on mortality outcome in two historical cohorts, 2019 (with exposure period ranging from February 1, 2019 to September 30, 2019 and outcome period ranging from February 1, 2019 to January 31, 2020) and 2018 (with exposure period ranging from February 1, 2018 to September 30, 2018 and outcome period ranging from February 1, 2018 to January 31, 2019). We excluded the telephone-only cohort in these historical cohort analyses, since telephone-only visits in pre-COVID times were not representative of medical visits, with only 13% of telephone calls lasting over 20 minutes in the pre-pandemic times, compared to 35% of telephone calls during the pandemic based on the current data.

## Statistical analyses

**Comparison of monthly outpatient visits and mortality rates between pre- and pandemic periods.** Mean monthly outpatient visit and crude mortality rates were plotted for two pre-COVID periods: February 1, 2018 to January 31, 2019 and February 1, 2019 to January 31, 2020 to serve as the baseline reference, to be compared against the COVID pandemic period (February 1, 2020-January 31, 2021); mean monthly visit and crude mortality rates between the pre-COVID periods were compared against the COVID period using T-tests. To determine whether a change over time occurred in monthly crude mortality rates, we used an interrupted time series analysis (ITSA) to compare the change over time (slope change), if any, in the crude monthly mortality rates between the pre-COVID periods (2/2018–1/2020) and the COVID period (2/2020–1/2021). To account for Veterans who contributed to multiple years of data and the case-mix of comorbidities within the study population, we conducted a population-averaged GEE logistic model and estimated marginal effects to determine if the mortality risk differed during the COVID period (2/2020–1/2021) compared to the pre-COVID periods (2/2018–1/2019 and 2/2019–1/2020) respectively. In this GEE model, the outcome was death, the COVID and pre-COVID periods were the fixed-effects, along with demographic and clinical covariates (described below), with the subject-specific intercept as the random-effects to account for repeated contributions to the data in multiple years for the same Veteran.

**Relationship between type-of-outpatient visit and mortality.** The index date is the first date of the exposure period (February 1, 2020) and patients were followed for all-cause mortality until May 31, 2021. Baseline characteristics at the time of the index date were compared among patients with no visits, with at least 1 in-person visit, with at least 1 video-only visit and with telephone-only visits. Chi-square analyses were used for categorical and ANOVA for continuous variables. We plotted Kaplan-Meier survival curves of HF patient groups stratified by presence of clinic days with 1) in-person, 2) video-only, or 3) telephone-only encounters, versus 4) the group with no-outpatient encounters. We related types of clinic encounters with mortality using Cox-regression models adjusting for potential demographic and clinical confounders obtained by cross-linkage of data between different databases within the VHA CDW (Corporate data warehouse). Baseline demographic covariates included were age, sex, race, and marital status at the start of the exposure period. Clinical covariates comprised of duration of heart failure diagnosis, comorbidities from the Elixhauser comorbidity index determined using ICD-10 codes, most recent (to February 1, 2020) left-ventricular-ejection-fraction (LVEF, categorized as <40%, ≥40%, or missing), brain natriuretic peptide (BNP) or n-terminal pro-brain natriuretic peptide (NT-proBNP) levels, positivity of COVID-19 test (by PCR or antigen test) and VHA healthcare use within one-year from the baseline (includes number of emergency-department visits, hospitalizations and hospice or palliative care visits) up to February 1, 2020. These covariates were chosen since they were known independent risk factors for mortality or potentially associated with presence and type of outpatient visits and mortality in HF.

We conducted sensitivity analyses by repeating our analysis relating type of outpatient visit (in-person, video-only, none) with mortality in historical cohorts of 2018 and 2019. We also assessed whether the specialty of the visit mattered by only including outpatient clinical encounters specifically with primary care or cardiology providers versus no visit with primary care or cardiology. In order to allow for adequate exposure time prior to outcome accrual, we also repeated our analyses using a blanking period, where death was only ascertained from October 1, 2020, onward. We further conducted subgroup analyses comparing mortality of patients with in-person encounters to those with video- or telephone-only encounters, excluding the no-outpatient visit group.

**Medication adherence during the pandemic.** To understand the potential mechanisms that may mediate the relationship between clinical encounters and mortality, we also compared medication possession ratio of commonly used cardiovascular medications (categorized by loop diuretics, angiotensin-converting-enzyme inhibitors, angiotensin-receptor blockers, angiotensin-receptor-neprilysin inhibitors, mineralocorticoid-receptor antagonists, beta-blockers, statins) amongst patients with in-person, video-only or telephone-only encounters versus none using ANOVA. Medication possession ratio was calculated as the number of days for which prescribed medication was available divided by the number of days during the observation period [12,13].

All analyses were performed using SAS Enterprise Guide 7.1 (SAS Institute Inc, Cary, NC). P-value <0.05 was considered statistically significant for the main analysis, and p-value <0.0167 was considered significant for subgroup analyses to account for multiple testing.

## Results

### Monthly rates in outpatient healthcare utilization between pre-pandemic and pandemic periods

Amongst n = 509,511 patients diagnosed with HF and cared for at the VHA between 2/2016–1/2020, the monthly outpatient visit-days were on average 81.4 (SD 6.1) visit-days in 2018–2019 and 81.0 (SD 5.6) visit-days in 2019–2020, which decreased to 57.8 (11) visit-days in 2020–2021 (P < 0.01 for comparison against both pre-COVID periods), per 100 Veterans. Conversely, monthly video-only visit days increased from a mean of 3.1 (SD 0.2) visit-days in 2018–2019 and 3.7 (SD 0.4) visit-days in 2019–2020, to 10.6 (3.6) visit-days in 2020–2021 (P < 0.01 for comparison against both pre-COVID periods), per 100 Veterans. As depicted in **Fig 1**, despite a significant increase in video-only visits, there were markedly

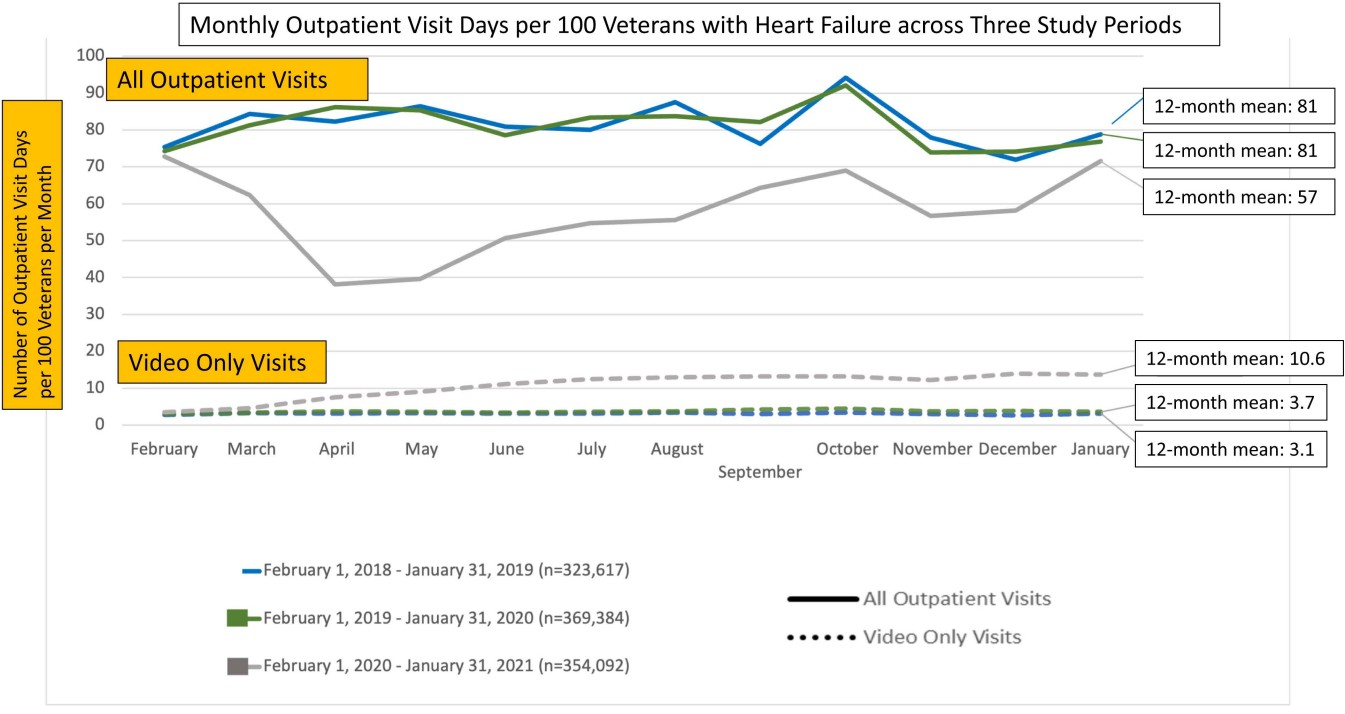

**Fig 1. Monthly outpatient visit days per 100 veterans with heart failure across three study periods.**

fewer overall outpatient visit-days compared to 2018 and 2019. A similar pattern was observed when restricting our observation to cardiology and primary care visits (S1 Data), despite similar age, gender, and race distribution (S1 Data).

## Monthly mortality rates between pre-pandemic and pandemic periods

Amongst the same patients, the monthly unadjusted mortality rates were similar between study periods with an average monthly mortality of 12.9 (SD 0.8) in 2018–2019 and 12.8 (SD 0.8) in 2019–2020, versus 13.1 (1.6) in 2020–2021 (P > 0.60 for both comparisons of COVID vs. pre-COVID periods), per 1000 Veterans (Fig 2). However, time series analysis for temporal trends showed a significant uptrend in the slope (Fig 3) for monthly mortality rates between the pre-COVID periods (2/2018–1/2020) versus the 2/2020–1/2021 (COVID period, P < 0.01). Adjusted analysis using GEE modeling showed that for a Veteran with HF and similar demographic and comorbid conditions, they were more likely to die during the COVID period compared to the 2/2018–1/2019 and 2/2019–1/2020 periods by 3.2% (95% CI: 3.1 to 3.4%) and 1.7% (95% CI: 1.6 to 1.9%) respectively.

## In-person, video-only, telephone-only or no-encounters and mortality

**Baseline characteristics.** A total of 321,439 Veterans with a diagnosis of HF were included, with a mean age [SD] 73.5 [10.5] years, 96.7% male, 74.5% White and 18.3% Black; of which, 14,266 patients (4.4%) had a COVID-19 positive test within the VHA system. The majority (56.1%) of patients had an LVEF ≥40%, 29.5% had an LVEF <40% and the remainder had missing LVEF information. Compared to the rest, patients with any in-person encounters had a higher prevalence of comorbidities such as hypertension, diabetes, and chronic pulmonary disease (**Table 1**, S1 Data). Examination of the rurality of Veteran residence showed a relatively narrow range amongst the groups, where a high percentage of Veterans resided in the urban settings, ranging from 92.7% for Veterans with video-only visits, 93.3% for Veterans with no visits, to 94.1% for Veterans with in-person visits.

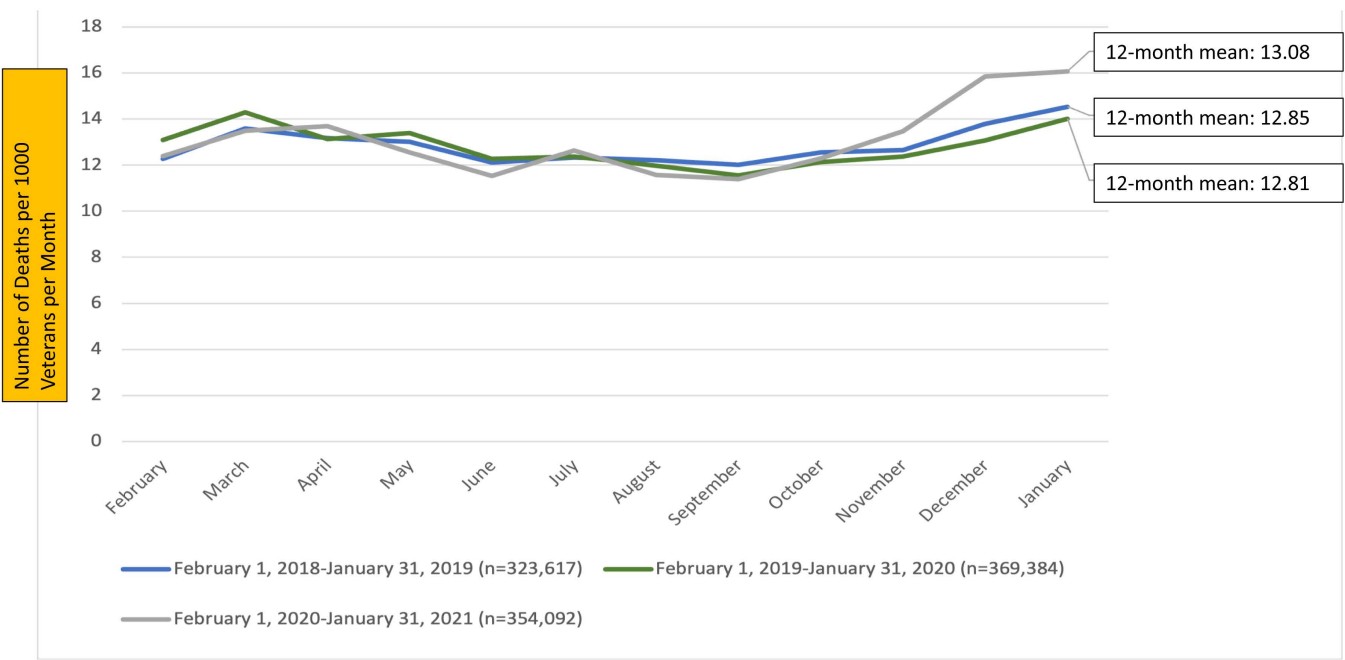

**Fig 2. Monthly mortality rates per 1000 veterans with heart failure across three study periods.**

**Mortality outcome by type of outpatient clinical encounter.** During the exposure period (2/1/2020–9/30/2020), 224,046 (69.7%) patients had in-person visits, 79,346 (24.7%) patients had video-only visits, 11,318 (3.5%) had telephone-only visits and 6,729 (2.1%) had no clinical visits. A total of 59,343 (18.5%) patients died in the outcome period (2/1/2020–5/31/2021). Kaplan-Meier curves for mortality stratified by the four groups are shown in **Fig 4**. Adjusted Cox-regression modeling showed that compared to patients with no outpatient visits, those with at least one day of in-person visits (HR 0.42, 95%CI: 0.41–0.44), at least one day of video-only visit (HR 0.52, 95%CI: 0.50–0.55), or at least one day of telephone-only visit (HR 0.57, 95%CI: 0.54–0.60) was associated with significantly lower mortality risks (p = 0.14 for linearity trend of the adjusted log hazards for mortality from in-person, video-only, to telephone-only visits) (**Table 2**). Sensitivity analyses in the historical cohorts (2/2018–1/2019 and 2/2019–1/2020), excluding telephone-only visits in the analyses, showed similar estimates and results (**S1 Data**). Subgroup analyses excluding the no-visit group, showed that compared to patients with telephone-only visits, patients with in-person visits (HR 0.75, 99%CI: 0.71–0.79) and those with video-only visits (HR 0.92, 99%CI: 0.87–0.97), were associated with lower mortality risks. With further stratification, those who had at least one day of in-person visits were associated with a lower mortality risk (HR 0.81; 99% CI: 0.79–0.83) compared to patients with only video visits. Sensitivity analyses which included a blanking period for outcome accrual (death accrual starting on 10/1/2020) showed similar findings (**S1 Data**). Sensitivity analyses that only included days with outpatient clinical encounters specifically within primary care or cardiology clinics also showed similar findings (**S1 Data**).

**Medication possession ratio during the pandemic.** The medication possession ratio was lowest amongst patients with no outpatient clinical encounters (mean adherence 75.7%), followed by those with telephone-only (mean adherence 83.9%) and those with virtual video (mean adherence 85%) encounters, and highest amongst those with in-person encounters (mean adherence 86.1%), p < 0.001.

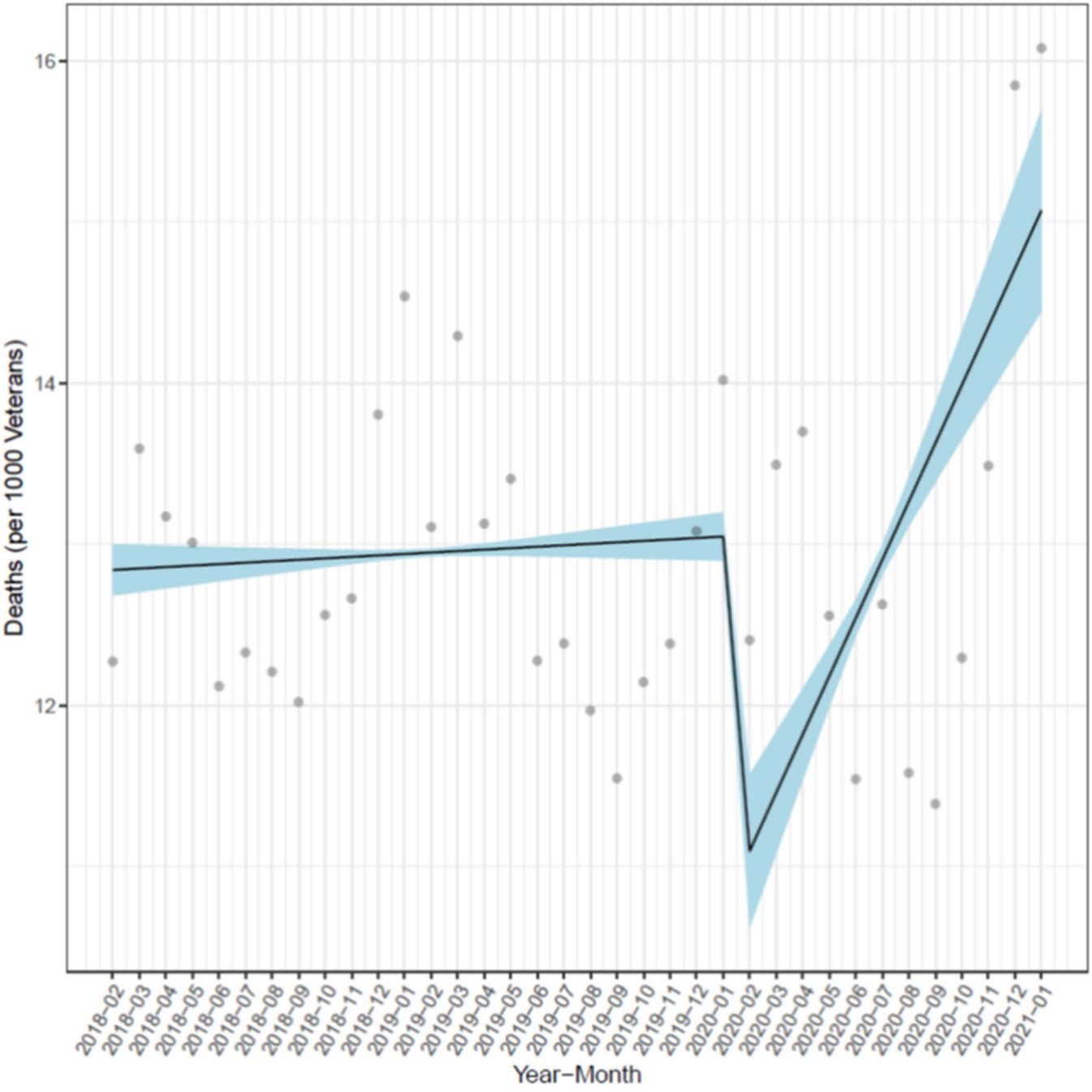

**Fig 3. Interrupted time series analysis of monthly mortality rates per 1000 veterans with heart failure during pre-COVID (2/2018-1/2020) and COVID pandemic (2/2020-1/2021) periods.**

**Table 1. Baseline characteristics of patients in the association between type of outpatient visit and subsequent mortality.**

| Characteristics, Mean (SD) or Number (%) | No Visit in 2020 | At least 1 in-person visit from February-September 2020 | At least 1 video-only visit from February-September 2020 | Telephone- only visit from February-September 2020 | Total |
|---|---|---|---|---|---|
| | n=6,729 | n=224,046 | n=79,346 | n=11,318 | n=321,439 |
| Age, years | 77.7 (12.2) | 73.3 (10.4) | 74.8 (10.9) | 77.1 (11.5) | 73.5 (10.5) |
| Gender | | | | | |
| Male | 6,512 (96.8%) | 216,229 (96.5%) | 77,042 (97.1%) | 11,055 (97.7%) | 310,838 (96.7%) |
| Race | | | | | |
| White | 4,966 (73.8%) | 165,654 (73.9%) | 60,139 (75.8%) | 8,659 (76.5%) | 239,418 (74.5%) |
| Black | 985 (14.6%) | 43,296 (19.3%) | 12,879 (16.2%) | 1,668 (14.7%) | 58,828 (18.3%) |
| Other | 167 (2.5%) | 4,886 (2.2%) | 1,835 (2.3%) | 223 (2.0%) | 7,111 (2.2%) |
| Missing | 611 (9.1%) | 10,210 (4.6%) | 4,493 (5.7%) | 768 (6.8%) | 16,082 (5.0%) |
| Married | 3,604 (53.6%) | 118,424 (52.9%) | 44,187 (55.7%) | 6,555 (57.9%) | 172,770 (53.7%) |
| Urban^ | 6,281 (93.3%) | 210,740 (94.1%) | 73,560 (92.7%) | 10,617 (93.8%) | 301,198 (93.7%) |
| VAMC Complexity^^ | | | | | |
| 1 | 5,460 (81.1%) | 188,850 (84.3%) | 63,798 (80.4%) | 9,385 (82.9%) | 267,493 (83.2%) |
| 2 | 591 (8.8%) | 16,199 (7.2%) | 6,499 (8.2%) | 913 (8.1%) | 24,202 (7.5%) |
| 3 | 678 (10.1%) | 18,997 (8.5%) | 9,049 (11.4%) | 1,020 (9.0%) | 29,744 (9.3%) |
| COVID-19 Positive Test within VA# | 60 (0.9%) | 11,498 (5.1%) | 2,535 (3.2%) | 173 (1.5%) | 14,266 (4.4%) |
| Left ventricular Ejection Fraction | | | | | |
| <40% | 1,759 (26.1%) | 66,207 (29.6%) | 23,415 (29.5%) | 3,292 (29.1%) | 94,673 (29.5%) |
| >=40% | 2,412 (35.8%) | 133,172 (59.4%) | 40,189 (50.7%) | 4,587 (40.5%) | 180,360 (56.1%) |
| Missing | 2,558 (38.0%) | 24,667 (11.0%) | 15,742 (19.8%) | 3,439 (30.4%) | 46,406 (14.4%) |
| BNP (pg/ml) | 948.5 (1420.7) | 686 (2751) | 640.8 (2879.1) | 621.4 (1162.4) | 679.1 (2728.7) |
| NT-proBNP (pg/ml) | 4,941.80 (8765.4) | 3,787.10 (9394.6) | 3,308.41 (6704.8) | 3,622.91 (6320.5) | 3,713.68 (8941.7) |
| Missing BNP or NTproBNP | 252 (3.7%) | 21,539 (9.6%) | 4,682 (5.9%) | 444 (3.9%) | 26,917 (8.4%) |
| Comorbidities^^^ | 252 (3.74%) | 21,539 (9.61%) | 4,682 (5.90%) | 444 (3.92%) | 26,917 (8.37%) |
| Cerebrovascular Disease | 451 (6.7%) | 26,778 (12.0%) | 7,030 (8.9%) | 737 (6.5%) | 34,996 (10.9%) |
| Diabetes | 1,955 (29.1%) | 116,741 (52.1%) | 34,833 (43.9%) | 3,783 (33.4%) | 157,312 (48.9%) |

*(Continued)*

**Table 1.** (Continued)

| Characteristics, Mean (SD) or Number (%) | No Visit in 2020 | At least 1 in-person visit from February-September 2020 | At least 1 video-only visit from February-September 2020 | Telephone- only visit from February-September 2020 | Total |
|---|---|---|---|---|---|
| | n = 6,729 | n = 224,046 | n = 79,346 | n = 11,318 | n = 321,439 |
| Hypertension | 3,633 (54.0%) | 180,217 (80.4%) | 57,109 (72.0%) | 7,142 (63.1%) | 248,101 (77.2%) |
| Chronic Pulmonary Disease | 1,194 (17.7%) | 74,985 (33.5%) | 20,939 (26.4%) | 2,233 (19.7%) | 99,351 (30.9%) |
| Obesity | 735 (10.9%) | 60,557 (27.0%) | 15,506 (19.5%) | 1,559 (13.8%) | 78,357 (24.4%) |
| Peripheral Vascular Disease | 652 (9.7%) | 43,256 (19.3%) | 10,907 (13.7%) | 1,097 (9.7%) | 55,912 (17.4%) |
| Pulmonary Circulation Disease[*] | 140 (2.1%) | 11,685 (5.2%) | 2,540 (3.2%) | 262 (2.3%) | 14,627 (4.6%) |
| Renal Failure, Moderate[*] | 742 (11.0%) | 37,230 (16.6%) | 10,719 (13.5%) | 1,306 (11.5%) | 49,997 (15.6%) |
| Renal Failure, Severe[*] | 193 (2.9%) | 14,911 (6.7%) | 3,241 (4.1%) | 275 (2.4%) | 18,620 (5.8%) |
| Valvular Heart Disease | 423 (6.3%) | 27,114 (12.1%) | 7,316 (9.2%) | 787 (7.0%) | 35,640 (11.1%) |

Chi-square analyses were used for categorical and ANOVA for continuous variables for between group comparison of all covariates, all P-values were < 0.001

^Urban setting determined using Rural-Urban Commuting Area (RUCA) codes from the Veterans' census tract, where codes 1–6 (Metropolitan area) represent urban setting.

^^The complexity VAMC model presented divides VHA facilities into 3 levels: level 1 represents high complexity, level 2 represents medium complexity, and level 3 represents low complexity, based on patient risk, levels of teaching/research activity, levels of ICU units, and number of Veterans Equitable Resource Allocation (VERA) pro-rated persons.

#COVID-19 positivity determined by either antigen or polymerase chain reaction (PCR) testing obtained within the VA system

^^All comorbidities defined by Elixhauser comorbidity definitions, determined using ICD10 codes from VA claims

*Determined by Elixhauser comorbidity definitions as:

1)Pulmonary Circulation Disease: primary pulmonary hypertension, atherosclerotic heart disease, other secondary pulmonary hypertension, unspecified pulmonary hypertension, secondary pulmonary arterial hypertension, pulmonary hypertension due to left heart disease, pulmonary hypertension due to lung diseases and hypoxia, chronic thromboembolic pulmonary hypertension, other secondary pulmonary hypertension, cor pulmonale (chronic), chronic pulmonary embolism, Eisenmenger's syndrome, other specified pulmonary heart diseases, unspecified pulmonary heart disease, arteriovenous fistula of pulmonary vessels, aneurysm of pulmonary artery, other diseases of pulmonary vessels, unspecified disease of pulmonary vessels

2)Renal Failure, Moderate: chronic kidney disease stage 3 (moderate), chronic kidney disease stage 3 unspecified, chronic kidney disease stage 3a, chronic kidney disease stage 3b, unspecified chronic kidney disease, unspecified kidney failure

3)Renal Failure, Severe: Hypertensive chronic kidney disease with stage 5 chronic kidney disease or end stage renal disease, Hypertensive heart and chronic kidney disease without heart failure (with stage 5 chronic kidney disease or end stage renal disease), Hypertensive heart and chronic kidney disease with heart failure (with stage 5 chronic kidney disease or end stage renal disease), chronic kidney disease stage 4 (severe), chronic kidney disease stage 5, end stage renal disease, encounter for fitting and adjustment of extracorporeal dialysis catheter, encounter for fitting and adjustment of peritoneal dialysis catheter, encounter for adequacy testing for hemodialysis, encounter for adequacy testing for peritoneal dialysis, patient's noncompliance with renal dialysis, kidney transplant status, dependence on renal dialysis

## Discussion

HF is a chronic, complex disease requiring close follow-up, and patients with HF represent a vulnerable population significantly affected by changes in provision and delivery of ambulatory healthcare during the pandemic. The effect of disruption and change in outpatient care due to clinic cancellations and social distancing policies early in the COVID pandemic for this vulnerable population was not known. Understanding the safety impact of these changes is critical in assessing optimal modes of HF care in the future. Our results show that despite the increase in video-only visits, there is still a net

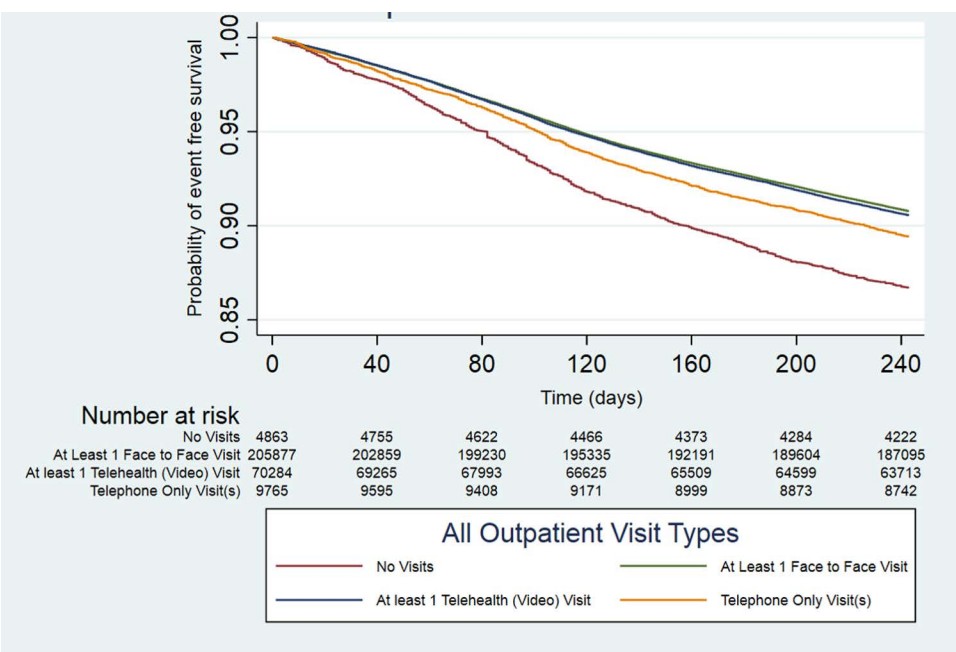

**Fig 4.** Kaplan-Meier curves during 2/1/2020-5/31/2021 among veterans with heart failure who had no outpatient visits, video-only visits, telephone-only visits and at least one face-to-face visit.

**Table 2. Association between type of outpatient visit and subsequent mortality.**

| DAYS WITH ANY OUTPATIENT VISIT COMPARED WITH NO VISIT (n = 321,439) | |
|---|---|
| Exposure in Model | Hazard Ratio (95% CI) |
| Any outpatient visit day vs. no visit | 0.47 (0.45, 0.49) |
| **TYPE OF VISIT (IN-PERSON, VIDEO-ONLY, OR TELEPHONE-ONLY) COMPARED WITH NO VISIT (n = 321,439)** | |
| Exposure in Model | Hazard Ratio (95% CI) |
| At least one in-person visit day vs. no visit | 0.42 (0.41, 0.44) |
| At least one video-only visit day vs. no visit | 0.52 (0.50, 0.55) |
| Telephone-only visit day vs. no visit | 0.57 (0.54, 0.60) |
| **AT LEAST ONE IN-PERSON VISIT DAY COMPARED WITH VIDEO-ONLY VISIT DAYS (n = 303,392)** | |
| Exposure in Model | Hazard Ratio (99% CI) |
| At least one in-person visit day vs. video-only visit day | 0.81 (0.79, 0.83) |
| **IN-PERSON AND VIDEO-ONLY VISIT DAYS COMPARED WITH TELEPHONE-ONLY VISIT DAYS (n = 314,710)** | |
| Exposure in Model | Hazard Ratio (99% CI) |
| At least one in-person visit day vs. telephone-only visit day | 0.75 (0.71, 0.79) |
| At least one video-only visit day vs. telephone-only visit day | 0.92 (0.87, 0.97) |

Cox regression adjusted for age, sex, race, marital status, ejection fraction, BNP/NT-pro-BNP, COVID positivity, number of visits in prior temporal time (February 2019-September 2019), hospitalization, ED visits, Elixhauser comorbidities, and hospital as fixed effects. P-values for all ≤ 0.001

decrease in outpatient clinic visits in 2/2020–1/2021 and suggest a significant disruption of care for patients with HF in the VHA. In a sub-sample of 321,439 patients who had at least one cardiology or primary care visit in 2019, mortality risk was associated with the presence and modality of outpatient visit, it was lowest for patients with at least one in-person visit, followed closely by video-only, then telephone-only visits, and highest for patients with no outpatient visits. This same pattern

of lower mortality risk associated with in-person and video-only visits compared to those with no outpatient visits was also observed in the pre-pandemic period (2018 and 2019).

The present study is, to our best knowledge, the first to assess disruption and change in care for HF patients during the COVID-19 pandemic and its association with mortality, using a nation-wide, large-scale patient data. Although overall death rate had increased in the US during COVID-19 in 2020 [14], study of excess mortality in the VA population shows that the most significant increase happened in the 4th quarter of 2020 [15], a pattern also shown in our study. Like other respiratory illnesses, COVID-19 has shown a tendency to decrease during summertime and increase again in fall and winter times. Further, we speculate that higher risk groups such as those with HF, may have been strictly isolated in the initial phase of the pandemic and may not have had significant exposure till later in the year. Despite the rise in mortality observed in the general US population related to the COVID-19 pandemic, the low (<5%) COVID-19 positivity rates in our study population of Veterans with HF suggest that COVID-19 infection may not have played a significant impact in the mortality risk of our population during the study period. Indeed, the current results suggest that the outpatient care changes, specifically, the type of HF visits and loss of visits, may partially explain the uptrend of mortality seen among Veterans with HF during the COVID-19 pandemic in the study.

Previous studies have shown that virtual visits allow patients to continue receiving medical care (including evaluation of clinical status, medication management and counseling) while reducing in-person exposure to the severe acute respiratory syndrome coronavirus-2 (SARS-CoV-2) [3,16]. In patients with long-term conditions requiring close follow-up, particularly patients with HF, cancer or diabetes mellitus, virtual visits have been shown to be a feasible option for patients to remain engaged and connected with the healthcare system [17,18], paving the way for these visits to potentially be a component of outpatient care in the future. Our study showed that both in-person and video-only visits were associated with decreased mortality risks compared to no-visits, with similar trends seen in the pre-pandemic analyses. Unlike prior reports of smaller sample size, confined within a certain region or using historical controls [19–22], our nationwide cohort and large sample size allowed for contemporaneous parallel comparisons of different outpatient visit modalities and obtained mortality risk estimates for each clinic modality. The distribution of baseline characteristics across visit type subgroups suggested that patients with in-person encounters were sicker, as evidenced by a higher prevalence of comorbidities such as hypertension, diabetes, and chronic pulmonary disease, among other conditions compared to the rest, which would have put them at a higher risk of mortality. Conversely, we found the medication-possession-ratio for cardiovascular medications, to be highest for patients with in-person, followed by video-only, then telephone-only and lowest amongst patients with no-outpatient visits. Therefore, we hypothesize that the lower mortality risks associated with patients with in-person visits may be related to the adherence to medications as a potential causal pathway to the association between type of encounter and mortality risk in the study population. A prior report evaluating cardiology ambulatory visits for HF have found a higher likelihood of guideline-directed medical therapy prescription [19] during in-person visits compared to remote visits for HF, video or telephone. It is reasonable to assume that providers were less likely to prescribe guideline-directed medical therapy when they were unable to examine the patient. Given the minimal 1% absolute difference in medication-possession-ratio between in-person and video-only visits in our report, the ability to visualize the patient via-video appeared to help with the continuous prescription of HF medications. Additionally, medication-possession-ratio may also be a marker for the patient's overall adherence to healthcare interventions and the results would suggest that those HF patients who were more adherent to health interventions were associated with better mortality outcomes.

It is important to note that for virtual visits to be successful, patients need both access to technology and skill and understanding of technology [3]. In the subgroup analyses, our data showed an association of a lower mortality risk for the in-person visits compared to video-only visits, which in turn, were associated with a lower mortality risk when compared to telephone-only visits. These findings are suggestive that although virtual visits can safely be implemented in patients with HF to provide continued access to care, they are likely not a complete substitute for in-person visits for HF. Future

randomized-controlled studies are needed to causally evaluate whether different encounter types to provide care in HF patients affect their mortality outcomes.

The decline in attendance to outpatient visits during COVID pandemic was not limited to the VA nor to HF patients but extended throughout the US, including other lifesaving cardiac interventions, such as cardiac rehabilitation [23]. This decline has led to closure of cardiac rehabilitation centers nationwide, a trend that persisted throughout the pandemic through September of 2022 [24]. Like our findings, the nationwide cardiac rehabilitation visit's decline was insufficiently compensated by the utilization of virtual cardiac rehabilitation. As such, the availability of a virtual option in care inter-spersed to the in-person visits may be an option to maintain access to services during epidemic surges of respiratory illness or to overcome other barriers in care such as geographic distance or transportation difficulty. In our study, we found Veterans with no-visits and using video-only visits were more likely to have a rural residence, which support the importance of virtual visits to maintain medical access for rural Veterans. Nonetheless, the range of rurality across the subgroups stratified by the type of visit was relatively narrow, within 1–2% point differences. This phenomenon is likely explained by the VHA policy that if the Veteran resides in an area too distant to a VHA clinic, they would likely be eligible for the VHA to pay for their medical care in the local community near their residence and would not have been included in the current study cohort. The associations found in the current study may provide the basis for the design of future trials to proactively re-design and optimize care modalities for the HF and other vulnerable patient populations, such as those with severe chronic obstructive pulmonary disease, cancer, diabetes mellitus on insulin-pump, among others.

## Study limitations

We acknowledge several limitations. First, given the observational nature of our study, there is a possibility that the observed associations are in part explained by unmeasured and/or inadequately measured confounding variables. The differential distribution in age, race, and important comorbidities such as diabetes and COPD, and type of HF, observed between the different visit groups may contribute to the difference in mortality between the groups. We attempted to account for these confounders by performing extensive adjustment for demographic variables including marital status, comorbidity burden and health care utilization variables, characteristics of the HF such as BNP and LVEF, rurality and COVID-19 test status. However, we acknowledge that residual confounding may persist and warrant consideration in interpreting the results. Factors such as perceived health status, ease of access to clinic or technology, and health lit-eracy can potentially impact the type of outpatient visits and subsequent mortality but are not adequately captured. Randomized-controlled studies are needed to confirm the causal impact between the visit types and clinical outcomes. Of note, the VHA has taken systemic measures to minimize transportation and digital disparities, which may reduce some of the confounding related to ability to use technology. For example, a "digital divide" consultation was established by the VHA to provide digital device, mobile applications, and health-related internet access for the patients, and/or education in the utilization of these devices to bridge the access gap in telehealth services. Patients who qualify for this consultation include those living >30 miles from a VHA medical facility, those with unstable housing, those with difficulty accessing public transportation or with conflicting commitments which make in-person visits challenging. Similar criteria would also apply for the VHA to provide transportation to clinical appointments for these patients. Despite the above, these programs often face numerous challenges, such as patient acceptance or the availability of sufficient institutional resources, which may limit its impact on the health-related digital divide. Second, while we studied a large, national cohort of patients, our patient sample was predominantly male and generalizability to female patients may be limited. Third, the outpatient visits analyzed were the ones reported by the VHA system, and clinic visits outside of the VHA system were not accounted for, which could have introduced a misclassification in the no-clinical visit group and potentially biased the mortality risk estimates toward the null-hypothesis. Additionally, healthcare systems other than the VHA may be organized differently, and generalization to other populations would need further investigation. Fourth, the COVID-19 testing analyzed were performed at the VA system and may have underestimated the total COVID-19 positivity of our cohort. Nonetheless, our

findings offered an insight into the impact of disruption in care for HF patients in the VHA system and potential opportunities for telehealth care models to bridge the gap in care.

## Conclusions

Despite an increase in video and use of telephone visits during the COVID-19 pandemic, there was still a decrease in total outpatient visits for patients with HF. The presence and type of outpatient encounter was associated with the adjusted risk of mortality. Future studies are needed to establish a causal relationship between visit modalities and mortality for patients with HF.

## Supporting information

**S1 Data. Supplement 1: Types of Clinic Visits Included in the Analysis Identified via Primary Clinic Stop Codes; Supplement 2: Primary Care/Cardiology Clinic Visits; Supplement 3: Virtual Video Visits and Telephone Visits; Supplement 4: Annual Cardiology and Primary Care Specific Outpatient Visit Days (either in-person or video-only) per 100 Veterans with Heart Failure Broken Down by Month from 2018-2020; Supplement 5: Annual Cardiology and Primary Care Specific Video-Only Outpatient Visit Days per 100 Veterans with Heart Failure Broken Down by Month from 2018-2020; Supplement 6: Number and characteristics of Veterans per year for Analysis of Patterns of Annual Outpatient Visits per Veteran and Deaths per 1000 Veterans per year for Veterans with Heart Failure between 2018 and 2020; Supplement 7: Complete List of Baseline Characteristics (with list of comorbidities by Elixhauser comorbidity definitions) for the Association of Type of Outpatient Visits with Subsequent Mortality in Patients with Heart Failure; Supplement 8: Association of Type of Outpatient Visits with Subsequent Mortality in Patients with Heart Failure Using Historical Cohorts (2018 and 2019); Supplement 9: Sensitivity Analyses for the Association of Type of Outpatient Visits with Subsequent Mortality in Patients with Heart Failure Using a Blanking Period; Supplement 10: Sensitivity Analyses for the Association of Type of Outpatient Visits with Subsequent Mortality in Patients with Heart Failure Including Days with Only Primary Care or Cardiology Outpatient Clinical Encounters.**
(DOCX)

## Acknowledgments

Drs. Vijayakumar, Erqou, Wu and Ms. Corneau contributed to conception and design, acquisition of data, analysis of data, and drafting and revising the manuscript. Dr. Kokkirala contributed to revising the manuscript for important intellectual content. All authors fulfill authorship criteria and have approved this manuscript. Views expressed here are those of the authors and do not represent the Department of Veterans Affairs.

## Author contributions

**Conceptualization:** Shilpa Vijayakumar, Emily Corneau, Sebhat Erqou, Wen-Chih Wu.

**Data curation:** Shilpa Vijayakumar, Emily Corneau, Sebhat Erqou, Wen-Chih Wu.

**Formal analysis:** Shilpa Vijayakumar, Emily Corneau, Sebhat Erqou, Wen-Chih Wu.

**Funding acquisition:** Wen-Chih Wu.

**Investigation:** Shilpa Vijayakumar, Emily Corneau, Sebhat Erqou, Wen-Chih Wu.

**Methodology:** Shilpa Vijayakumar, Emily Corneau, Sebhat Erqou, Aravind Kokkirala, Wen-Chih Wu.

**Project administration:** Wen-Chih Wu.

**Resources:** Emily Corneau, Wen-Chih Wu.

**Software:** Wen-Chih Wu.

**Supervision:** Wen-Chih Wu.

**Visualization:** Shilpa Vijayakumar, Emily Corneau, Sebhat Erqou, Wen-Chih Wu.

**Writing – original draft:** Shilpa Vijayakumar, Emily Corneau, Sebhat Erqou, Wen-Chih Wu.

**Writing – review & editing:** Shilpa Vijayakumar, Emily Corneau, Sebhat Erqou, Aravind Kokkirala, Wen-Chih Wu.

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
