## [Decision Letter · Decision Letter 0]

21 Jun 2024

PONE-D-24-06893Outpatient care changes and associated mortality among Veterans with heart failure during the COVID-19 pandemicPLOS ONE

Dear Dr. Wu, Thank you for submitting your manuscript to PLOS ONE. After careful consideration, we feel that it has merit but does not fully meet PLOS ONE’s publication criteria as it currently stands. Therefore, we invite you to submit a revised version of the manuscript that addresses the points raised during the review process. The reviewer's comments may be found at the end of this email.

We look forward to receiving your revised manuscript.

Kind regards,

Pracheth Raghuveer, MD, DNB

Academic Editor

PLOS ONE

Journal Requirements:

2. Thank you for stating the following financial disclosure: "Department of Veterans Affairs, HSRD funding IRP 20-003".

3. Thank you for stating the following in the Competing Interests section: "Dr. Wu and Emily Corneau have no conflicts to declare. Dr. Vijayakumar is supported by NHLBI T32 postdoctoral training grant T32HL094201and a research grant from the Amyloidosis Foundation. Dr. Erqou is supported by the Department of Veterans Affairs, VISN-1 Career Development Award. Dr. Kokkirala owns shares in TDOC and AMWL telehealth stocks. There are no relationships with industry."

Reviewers' comments:

Reviewer's Responses to Questions

**Comments to the Author**

1. Is the manuscript technically sound, and do the data support the conclusions?

Reviewer #1: Yes

Reviewer #2: Yes

Reviewer #3: Partly

2. Has the statistical analysis been performed appropriately and rigorously? 

Reviewer #1: Yes

Reviewer #2: Yes

Reviewer #3: Yes

3. Have the authors made all data underlying the findings in their manuscript fully available?

Reviewer #1: No

Reviewer #2: No

Reviewer #3: Yes

4. Is the manuscript presented in an intelligible fashion and written in standard English?

Reviewer #1: Yes

Reviewer #2: Yes

Reviewer #3: Yes

5. Review Comments to the Author

Reviewer #1: This national study represents outcome data for approximately 500,000 (predominantly self-identified) male patients diagnosed with Heart Failure (HF) within the Veterans Health Adm (VHA). The study compares mortality outcomes for two different "pre-COVID" time periods versus the Febr 1, 2020 to Jan 31, 2021 "Covid period." The main exposure variable is type of outpatient visit (none; in-person; video; or phone). The study controls for a broad set of confounders/co-variates and applies advanced statistics incl General Estimation Equation, in the analysis. For a subset, COX regression modeling is used, as well as Kaplan Meier Survival Analysis.

The main study findings are: Decreased outpatient visits for HF during Covid vs pre-Covid; Highest risk-adjusted mortality rate among the "non-visit" group followed by phone visits; video consultation, and lowest for those with at least one outpatient, in-person visit.

The study also explores pharma adherence (possession) as a possible mediator between visit type and mortality outcomes. The authors recognize several study limitations, including observational nature. They recommend possible next steps to strengthen study reliability including randomized clinical trials. The authors also discuss possible clinical implications.

Specific Comments:

Table 1. In the legend, provide more detailed info re stat applied to determine, for example, stat differences across groups in pulmonary Vascular Disease. Legend only states P <....

Table 2. Excellent summary of the COX regression modeling.

Figure 1. Difficult to delineate which line represents Total Outpatient Visits Febr/2020-ff.

Figure 2. Discuss possible reasons that the apparent increase in mortality during Covid did not occur until November and forward of 2020.

Expand discussions re distribution of established risk factors, e g BNP, across visit groups and possible impact on survival, in addition to pharma adherence. The Cox modeling adjusts for a large set of risk factors/co-variates, but there is no discussion as to their possible pathophysiological role of such risk factors/covariates in mediating/moderating the mortality outcome. It's especially interesting that there are no stat sign differences in biomarkers (although in dx) across visit groups in Table 1. Several of these biomarkers are closely related to HF outcomes, e.g., BNP.

Expand discussion re implication of decreased overall outpatient visits (although virtual (video and phone) visits have increase) during Covid. Is this a trend that holds true still? I don't request additional analysis. However, the authors mention the need for further studies of the optimal mix of in-person vs virtual visits. There is no data presented, as far as this reviewer can find, that outcomes are better than pre-Covid for any of the three visit types. When comparing mortality rates over time, how does rate look when only adjusting for pre vs peri Covid periods? Did VHA have sufficient virtual (video and phone) visits pre-Covid to allow for meaningful analysis. Expand discussions as to the drivers of more virtual visits during Covid. Are there any indications that the most (multi) sick patients, and withe the worst LVEF were more likely to avoid in-person healthcare visits? Table 1 does not show significant differences across several of the categories, e.g., urban vs rural that the authors discuss the VHA has addressed existing digital divide. Rather than merely listing the various programs the VHA has launched to address socioeconomic/racial and geographic digital divide, the authors should recognize the challenges with these programs to actually impact health-related divide.

Reviewer #2: I have provided my comments as an attachment to the authors as well.

Thank you for the opportunity to review this paper. Understanding how the shift to telehealth during COVID in the VA affected mortality is very important, particularly in the context of potential policy changes being debated at CMS. I have some comments for the authors below.

Major comments:

1. Please modify odds ratios to be marginal effects. See a recent publication Norton et al. on the challenges with interpreting odds ratios outside of case-control studies: https://onlinelibrary.wiley.com/doi/10.1111/1475-6773.14337

2. The authors write that “the main exposure was outpatient visit day…” I’m not sure what this means. Could the authors elaborate on how the exposure was calculated? Is it a binary variable indicator whether the patient had an outpatient visit at any time in the outcome period? Is it the number of visits? What level of observation is used here?

3. I have some concerns about the statistical analysis that the authors conducted

a. It would be helpful to see an estimating equation. What is the outcome? What is on the right-hand side? It looks like the authors used both a Cox regression and a GEE? What is the reason for estimating both? Why would time to event be a useful measure here?

b. It looks like the authors included VA health care use for one year from the baseline. Can the authors explain a little more what this looked like? Are they summing all visits before the pandemic period? A little clarification would be helpful. I’m hoping to verify that the authors did not include utilization during the pandemic period.

c. How was the variable for positive COVID-19 test coded? Was that a binary indicator? Additionally, where was this information obtained? From VHA CDW?

d. The authors write that they tested for effect modification by COVID-19 status and several other variables by fitting interaction terms. Because there are issues with interaction terms in non-linear models, these are difficult to interpret. I suggest the authors estimate these as stratified models instead.

4. I’m glad to see that the authors ran their models in 2018 and 2019 as well. Those results, in the appendix now, should be discussed somewhat in the context of their findings. The fact that there is a mortality difference in 2020 between in-person and other visit types seems consistent with the pattern in 2018 and 2019. Are there possibilities for patient selection? For instance, are high-need patients in rural areas with less appointment availability more likely to use telehealth or not receive a visit to begin with?

Reviewer #3: The authors  have produced an analysis of the association of visit modality and HF mortality prior to and during the COVID-19 pandemic. There are several highly commendable aspects of this manuscript, including the comprehensive detailing of the methods and a clear and interesting discussion. The inclusion of the medication possession ratio analysis to begin to explore one possible mechanism is a strength. I do have some suggestions for improvement:

Major:

My biggest concern is regarding the study design and the conclusions it allows one to draw; there are likely a number of unobservable confounders when comparing individuals who had no in-person or no video-based care over the exposure period to those who did, including frailty, beliefs about healthcare/the pandemic, and general engagement in care (particularly when comparing to those with no follow-up). This is noted in the limitations, but I worry that it significantly limits interpretation of these findings. In general, while the authors do this to some extent, it is important to avoid drawing causal conclusions from these results.

What do you make of the fact that the same pattern of lower mortality risk with in-person and video-only visits compared to those with no outpatient visits was also observed in the pre-pandemic period (2018 and 2019)? A bit more discussion/interpretation of these findings would be helpful.

Minor:

It would be helpful to the reader in the abstract and conclusion to note that while there was an increase in virtual visits as stated, this was only partially compensatory (i.e., overall follow-up was decreased during the pandemic compared to pre-pandemic period).

I may be misinterpreting, but for all the figures with visits per 100 Veterans, if the Y-axis is showing annual visit rates per 100 Veterans--wouldn't that mean that e.g., an annual rate of 80 visits per 100 Veterans = 0.8 annual visits, not 8?

With regard to your statement that "the low (<5%) COVID-19 positivity rates in our study population suggest that COVID-19 infection may not have played an important role in the higher annual mortality rate seen in 2020"--given you are only capturing COVID-19 tests done in the VA system and the general difficulty of obtaining a test at that point in the pandemic, it seems likely that the actual rate of COVID-19 positivity was much higher than this (as you note in the limitations), which undermines your conclusion that disruption in outpatient care is what caused the increased mortality. Very minor, but in the following sentence, "Our results suggest that the disruption and change in outpatient care maybe related to the increase mortality in this vulnerable population", there should be a space between "may" and "be" and I believe it should say "increased" mortality with a "d".

On p. 15, "sever" should be "severe".

It would be helpful to understand why you defined 1/31/2020 as the end of the pre-pandemic period, as most lock-downs and disruptions in care began a bit later.

The supplement states that "Telephone visits are not specified as cardiology/primary care"--in fact they are, but require accounting for both primary and secondary stop codes.

6. PLOS authors have the option to publish the peer review history of their article (what does this mean? ). If published, this will include your full peer review and any attached files.

**Do you want your identity to be public for this peer review?** For information about this choice, including consent withdrawal, please see our Privacy Policy .

Reviewer #1: **Yes: ** Bengt B. Arnetz, M.D., PhD

Reviewer #2: No

Reviewer #3: No

---

## [Author Response · Author response to Decision Letter 1]

5 Sep 2024

August 26th, 2024

Rebuttal letter

Dear Members of the Editorial Staff & Reviewers, PLoS One,

We are pleased to re-submit our manuscript entitled, “Outpatient care changes and associated mortality among Veterans with heart failure during the COVID-19 pandemic” for consideration for publication in PLoS One. We have carefully reviewed all comments made by the Editorial Board and Reviewers and revised our manuscript accordingly. Please see below for responses to the comments and suggestions made by the Editorial Board and Reviewers. All suggested changes have been made in the re-submitted manuscript.

Reviewer #1:

1) Table 1. In the legend, provide more detailed info re stat applied to determine, for example, stat differences across groups in pulmonary Vascular Disease. Legend only states P <....

We apologize for the oversight. We have added the stats used for the comparison of baseline characteristics under the Statistical Analyses section of the Methods and in the footnote for Table 1: “Chi-square analyses were used for categorical and ANOVA for continuous variables.”

2) Table 2. Excellent summary of the COX regression modeling.

Thank you!!

3) Figure 1. Difficult to delineate which line represents Total Outpatient Visits Febr/2020-ff.

We have now added subheading labels for total outpatient visits and video only visits for the graph and included the figure on a separate file in a high quality figure.

4) Figure 2. Discuss possible reasons that the apparent increase in mortality during Covid did not occur until November and forward of 2020.

We thank the reviewer for noting this point. We have added in the second paragraph of the Discussion the potential reasons: “Although overall death rate had increased in the US during COVID-19 in 2020 [14], study of excess mortality in the VA population shows that the most significant increase happened in the 4th quarter of 2020 [15], a pattern also shown in our study. Like other respiratory illnesses, COVID-19 has shown a tendency to decrease during summertime and increase again in fall and winter times. Further, we speculate that higher risk groups such as those with HF, may have been strictly isolated in the initial phase of the pandemic and may not have had significant exposure till later in the year. Despite the rise in mortality observed in the general US population related to the COVID-19 pandemic, the low (<5%) COVID-19 positivity rates in our study population of Veterans with HF suggest that COVID-19 infection may not have played a significant impact in the mortality risk of our population during the study period. Indeed, despite a significant uptrend in the slope of monthly mortality rates in the 2/2020-1/2021 period for our population compared to previous years, the annual average in monthly-mortality rates compared to the previous 2 years remained similar.”

5) Expand discussions re distribution of established risk factors, e g BNP, across visit groups and possible impact on survival, in addition to pharma adherence.

We thank the reviewer for this comment. We have now added a new paragraph (Discussion, para 4) : “It is important to note that the distribution of baseline characteristics across visit groups suggested that patients with in-person encounters were sicker as evidenced by a higher prevalence of comorbidities such as hypertension, diabetes, and chronic pulmonary disease, among other conditions compared to the rest, which would have put them at a higher risk of mortality. Conversely, we found the medication-possession-ratio for cardiovascular medications, to be highest for patients with in-person, followed by video-only, then telephone-only and lowest amongst patients with no-outpatient visits. Therefore, it is reasonable to hypothesize that the lower mortality risk associated with the in-person visit patient group may be related to the adherence to medications as a potential causal pathway to the association between type of encounter and mortality risk in the study population.”

6) The Cox modeling adjusts for a large set of risk factors/co-variates, but there is no discussion as to their possible pathophysiological role of such risk factors/covariates in mediating/moderating the mortality outcome. It's especially interesting that there are no stat sign differences in biomarkers (although in dx) across visit groups in Table 1. Several of these biomarkers are closely related to HF outcomes, e.g., BNP.

We included in the methods “These covariates were chosen since they are known or expected independent risk factors for mortality and potentially associated with presence and type of outpatient visit days and mortality in HF.” Given the large sample size in each of the groups stratified by type of visits, the comparisons of the different baseline characteristics, including BNP levels, were statistically significant between the groups. This was included as part of the legend in the footnote of Table 1. We have also incorporated the potential impact of the comorbidity distribution in mortality risk estimation in the discussion paragraph 4 (see response to comment #4), and the measures we have taken to account for those factors within the limitation paragraph of the discussion.

7) Expand discussion re implication of decreased overall outpatient visits (although virtual (video and phone) visits have increase) during Covid. Is this a trend that holds true still? I don't request additional analysis. However, the authors mention the need for further studies of the optimal mix of in-person vs virtual visits. There is no data presented, as far as this reviewer can find, that outcomes are better than pre-Covid for any of the three visit types.

The reviewer is correct that no published data is available (to our knowledge) of the comparison of volume of in-person and virtual visits during COVID vs. pre-COVID times. Indeed, available data in other type of cardiac visits such as the provision of cardiac rehabilitation for Medicare beneficiaries suggest that the encounter volume for cardiac rehabilitation continue to be lower than pre-pandemic times. To discuss these points, we have expanded on the additional implication of the decreased overall outpatient visits (although virtual [video and phone] visits have increased) during COVID times in the non-VA sector using cardiac rehabilitation as an example. We added these sentences to the discussion and have deleted the assertion that the mix of in-person and virtual visits were optimal since there is no data to support such statement. The revised paragraph reads (last para of discussion prior to limitations): “the decline in attendance to outpatient visits was not limited to the VA but extended throughout the US, including other lifesaving cardiac interventions, such as cardiac rehabilitation. This decline has led to closure of cardiac rehabilitation centers, a trend that persisted throughout the pandemic through September of 2022. Like our findings, the cardiac rehabilitation visit’s decline was insufficiently compensated by the utilization of virtual cardiac rehabilitation visits. As such, the availability of a virtual option in care interspersed to the in-person visits may be an option to maintain access to services during epidemic surges of respiratory illness or to overcome other barriers in care such as geographic distance or transportation difficulty. These results may provide the basis for the design of future trials to proactively re-design and optimize care modalities for the HF and other vulnerable patient populations, such as those with severe chronic obstructive pulmonary disease, cancer, diabetes mellitus on insulin-pump, among others.”

8) When comparing mortality rates over time, how does rate look when only adjusting for pre vs peri Covid periods?

We have now formally compared the average monthly mortality rates between the pre-COVID periods and the COVID period using T-tests and added a time series analysis to compare the change in monthly mortality rates over time between the pre-COVID and COVID periods (under statistical analyses in methods): “mean monthly visit and mortality rates between the pre-COVID periods were compared against the COVID period using T-tests. To determine whether a change over time occurred in monthly mortality rates, we used an interrupted time series analysis (ITSA) to compare the slope change, if any, in the monthly mortality rates between the pre-COVID periods (1/2018-1/2020) and the COVID period (2/2020-1/2021).”

In results, we added (second para): “the monthly unadjusted mortality rates were similar between study periods with an average monthly mortality of 12.9 (SD 0.8) in 2018-2019 and12.8 (SD 0.8) in 2019-2020, versus 13.1 (1.6) in 2020-2021 (P>0.60 for both comparisons of COVID vs. pre-COVID periods), per 1000 Veterans (Figure 2). However, time series analysis for temporal trends showed a significant uptrend in the slope for monthly mortality rates between the pre-COVID periods (2/2018-1/2020) versus the 2/2020-1/2021 (COVID period, P<0.01).

9) Did VHA have sufficient virtual (video and phone) visits pre-Covid to allow for meaningful analysis.

We were able to obtain sufficient pre-COVID data for meaningful analysis of the video visits but not telephone visits, since in pre-COVID times, telephone calls alone were not considered a valid clinical visit in the VA.

We stated: “Sensitivity analyses in the historical cohorts (2/2018-1/2019 and 2/2019-1/2020), excluding telephone-only visits in the analyses, showed similar estimates and results (supplement 8). Supplemental Table 8 contains the analyses for video visits:

2/2019-1/2020 Analysis

TYPE OF VISIT (IN-PERSON or VIDEO-ONLY) COMPARED WITH NO VISIT (n=196,910)

Exposure in Model Hazard Ratio (95% CI)

At least one in-person visit day (n=173,899) vs. no visit (n=7,315) 0.63 (0.58, 0.68)

At least one video-only visit day (n=1,617) vs. no visit (n=7,315) 0.76 (0.64, 0.90)

2/2018-1/2019 Analysis

TYPE OF VISIT (IN-PERSON or VIDEO-ONLY) COMPARED WITH NO VISIT (n=161,118)

Exposure in Model Hazard Ratio (95% CI)

At least one in-person visit day (n=142,834) vs. no visit (n=4,935) 0.39 (0.25, 0.63)

At least one video-only visit day (n=1,169) vs. no visit (n=4,935) 0.48 (0.14, 1.63)

We included under “Historical Cohorts” section, in Methods the following: “We excluded the telephone-only cohort in these historical cohort analyses, since telephone-only visits in pre-COVID times were not representative of medical visits”.

10) Expand discussions as to the drivers of more virtual visits during Covid. Are there any indications that the most (multi) sick patients, and with the worst LVEF were more likely to avoid in-person healthcare visits? Table 1 does not show significant differences across several of the categories, e.g., urban vs rural that the authors discuss the VHA has addressed existing digital divide. Rather than merely listing the various programs the VHA has launched to address socioeconomic/racial and geographic digital divide, the authors should recognize the challenges with these programs to actually impact health-related divide.

We thank the reviewer for this important point. Per the response to previous comments #5 and #6, we have added in the discussion section how there was a higher prevalence of sicker patients in the in-person visit group (all differences were statistically significant per table footnote) and how multi-variate regression analyses were able to account for the differential distribution of comorbidities and yet still found the in-person (sicker) visit group to have a reduced mortality risk compared to the virtual- or no-visit groups. We added the following to the limitation section of the discussion: “The differential distribution in age, race, and important comorbidities such as diabetes and COPD, and type of HF, that is observed between the different visit groups may contribute to the difference in mortality between the groups. We attempted to account for these confounders by performing extensive adjustment for demographic variables including marital status, comorbidity burden and health care utilization variables, characteristics of the HF such as BNP and LVEF, rurality, and COVID-19 test status. However, we acknowledge that residual confounding may persist and warrant consideration in interpreting the results. Factors such as perceived health status, ease of access to clinic or technology, and health literacy can potentially impact the type of outpatient visits and subsequent mortality but are not adequately captured..”

In terms of the digital divide we also added: “It is also important to note that that the VHA has taken systemic measures to minimize transportation and digital disparities, which may reduce some of the confounding related to ability to use technology….Despite the above, these programs often face numerous challenges such as patient acceptance or the availability of sufficient institutional resources, which may limit its impact on the health-related digital divide.”

Reviewer #2:

1. Please modify odds ratios to be marginal effects. See a recent publication Norton et al. on the challenges with interpreting odds ratios outside of case-control studies: https://onlinelibrary.wiley.com/doi/10.1111/1475-6773.14337

We agree with the reviewer of this important observation of showing marginal effects in lieu of odds ratios. We also agree with the concern raised in comment #3 below on whether a GEE modeling is needed to compare temporal trends in mortality rates, which produced odds ratios and have made the interpretation of the results complex and difficult. Reviewer 1 (comment #8) also suggested an unadjusted analysis of the change in mortality rates over time between pre-covid and covid periods. As such, we have decided to exclude the GEE analysis from the paper. Instead, we added simple T-tests for comparison of average monthly mortality rates between the pre-COVID and COVID periods and a time series analysis to compare the change over time (slope), if any, in monthly mortality rates between the pre-COVID and COVID periods for easy interpretation. To this effect, we have deleted the GEE analysis from the methods and results (including odds ratios) and added the following (under statistical analyses in methods): “mean monthly visit and mortality rates between the pre-COVID periods were compared against the COVID period using T-tests. To determine whether a change over time occurred in monthly mortality rates, we used an interrupted time series analysis (ITSA) to compare the change over time (slope change), if any, in the monthly mortality rates between the pre-COVID periods (1/2018-1/2020) and the COVID period (2/2020-1/2021).”

2. The authors write that “the main exposure was outpatient visit day…” I’m not sure what this means. Could the authors elaborate on how the exposure was calculated? Is it a binary variable indicator whether the patient had an outpatient visit at any time in the outcome period? Is it the number of visits? What level of observation is used here?

We apologize for the confusion; “outpatient visit day” refers to the number of days with outpatient encounters, a continuous variable. We chose “days” as opposed to “encounters” as the unit of exposure because patients may have multiple clinical encounters and of different types within the same day. In cases where multiple clinical encounters of different types (e.g. in-person, video and telephone) occurred within the same day, we used the encounter with the highest intensity of contact (in-person > video-only > telephone-only) as the visit type for the day. We have added the above text in the Exposure section of the Methods to further clarify the unit of exposure.

3. I have some concerns about the statistical analysis that the authors conducted

a. It would be helpful to see an estimating equation. What is the outcome? What is on the right-hand side? It looks like the authors used both a Cox regression and a GEE? What is the reason for estimating both? Why would time to event be a useful measure here?

Please see answer to comment #1 above regarding deleting the GEE model and instead using an interrupted time series analysis to compare the change in monthly mortality rates between

---

## [Decision Letter · Decision Letter 1]

7 Oct 2024

PONE-D-24-06893R1Outpatient care changes and associated mortality among Veterans with heart failure during the COVID-19 pandemicPLOS ONE

Dear Dr. Wu,

Thank you for submitting your manuscript to PLOS ONE. After careful consideration, we feel that it has merit but does not fully meet PLOS ONE’s publication criteria as it currently stands. Therefore, we invite you to submit a revised version of the manuscript that addresses the points raised during the review process. 1) I'm not sure why the authors decided to remove the GEE estimates. I certainly didn't intend for that, instead, I would have removed the time-to-event modeling. I asked the authors initially why time-to-event would be an interesting outcome here, but they didn't respond to that question. I think the GEE answers the more useful question of the overall relationship with mortality rather than how much time passed before death. My suggestion is to remove the Cox Regression and Kaplan Meier curves, and focus on the results from the GEE with marginal effects.

2) The authors write that they performed interrupted time series analysis. It would be very helpful to see the interrupted time series figure (showing the slope change) to drive the point that they're making.

We look forward to receiving your revised manuscript.

Kind regards,

Pracheth Raghuveer, MD, DNB

Academic Editor

PLOS ONE

Journal Requirements:

Reviewers' comments:

Reviewer's Responses to Questions

**Comments to the Author**

1. If the authors have adequately addressed your comments raised in a previous round of review and you feel that this manuscript is now acceptable for publication, you may indicate that here to bypass the “Comments to the Author” section, enter your conflict of interest statement in the “Confidential to Editor” section, and submit your "Accept" recommendation.

Reviewer #1: All comments have been addressed

Reviewer #2: (No Response)

2. Is the manuscript technically sound, and do the data support the conclusions?

Reviewer #1: Yes

Reviewer #2: Yes

3. Has the statistical analysis been performed appropriately and rigorously? 

Reviewer #1: Yes

Reviewer #2: Yes

4. Have the authors made all data underlying the findings in their manuscript fully available?

Reviewer #1: Yes

Reviewer #2: No

5. Is the manuscript presented in an intelligible fashion and written in standard English?

Reviewer #1: Yes

Reviewer #2: Yes

6. Review Comments to the Author

Reviewer #1: The authors have provided acceptable and valid comments to all of this reviewer's critique. The revised manuscript is written in a way that provides responses to all of my initial critiques.

Reviewer #2: I appreciate the authors' responses to my comments.

I have a few additional suggestions/comments.

1) I'm not sure why the authors decided to remove the GEE estimates. I certainly didn't intend for that, instead, I would have removed the time-to-event modeling. I asked the authors initially why time-to-event would be an interesting outcome here, but they didn't respond to that question. I think the GEE answers the more useful question of the overall relationship with mortality rather than how much time passed before death. My suggestion is to remove the Cox Regression and Kaplan Meier curves, and focus on the results from the GEE with marginal effects.

2) The authors write that they performed interrupted time series analysis. It would be very helpful to see the interrupted time series figure (showing the slope change) to drive the point that they're making.

7. PLOS authors have the option to publish the peer review history of their article (what does this mean? ). If published, this will include your full peer review and any attached files.

**Do you want your identity to be public for this peer review?** For information about this choice, including consent withdrawal, please see our Privacy Policy .

Reviewer #1: **Yes: ** Bengt B. Arnetz

Reviewer #2: No

---

## [Author Response · Author response to Decision Letter 2]

15 Oct 2024

October 10th, 2024

Dear Members of the Editorial Staff & Reviewers, PLoS One,

We are pleased to re-submit our manuscript entitled, “Outpatient care changes and associated mortality among Veterans with heart failure during the COVID-19 pandemic” for consideration for publication in PLoS One. We have carefully reviewed all comments made by the Editorial Board and Reviewers and revised our manuscript accordingly. Please see below for responses to the comments and suggestions made by the Editorial Board and Reviewers. All suggested changes have been made in the re-submitted manuscript.

Reviewer:

1) I'm not sure why the authors decided to remove the GEE estimates. I certainly didn't intend for that, instead, I would have removed the time-to-event modeling. I asked the authors initially why time-to-event would be an interesting outcome here, but they didn't respond to that question. I think the GEE answers the more useful question of the overall relationship with mortality rather than how much time passed before death. My suggestion is to remove the Cox Regression and Kaplan Meier curves, and focus on the results from the GEE with marginal effects.

R/ We apologize for mis-interpreting the comments. We have re-integrated the GEE model to the paper and reported the marginal effects as opposed to ORs to study the risk-adjusted relationship between the impact of the COVID pandemic and mortality in patients with HF overall.

We would also like to keep the Cox Regression and Kaplan Meier curves as it is studying a different question, which is the risk-adjusted relationship between type-of-outpatient visits (in-person, video, telephone, none) and mortality in patients with heart failure.

2) The authors write that they performed interrupted time series analysis. It would be very helpful to see the interrupted time series figure (showing the slope change) to drive the point that they're making.

R/ We have now added the interrupted time series figure that shows a slope change in monthly crude mortality rates, between the pre-COVID periods (2/2018-1/2020) and the COVID period (2/2020-1/2021) as a new Figure 3.

On behalf of the co-authors, we sincerely thank the reviewers for their thoughtful insights which had made significant improvements to the paper.

Wen-Chih Wu, MD, MPH

---

## [Decision Letter · Decision Letter 2]

18 Feb 2025

PONE-D-24-06893R2Outpatient care changes and associated mortality among Veterans with heart failure during the COVID-19 pandemicPLOS ONE

Dear Dr. Wu,

Thank you for submitting your manuscript to PLOS ONE. After careful consideration, we feel that it has merit but does not fully meet PLOS ONE’s publication criteria as it currently stands. Therefore, we invite you to submit a revised version of the manuscript that addresses the points raised during the review process.

We look forward to receiving your revised manuscript.

Kind regards,

Pracheth Raghuveer, MD, DNB

Academic Editor

PLOS ONE

Journal Requirements:

Reviewers' comments:

Reviewer's Responses to Questions

**Comments to the Author**

1. If the authors have adequately addressed your comments raised in a previous round of review and you feel that this manuscript is now acceptable for publication, you may indicate that here to bypass the “Comments to the Author” section, enter your conflict of interest statement in the “Confidential to Editor” section, and submit your "Accept" recommendation.

Reviewer #1: All comments have been addressed

2. Is the manuscript technically sound, and do the data support the conclusions?

Reviewer #1: Yes

3. Has the statistical analysis been performed appropriately and rigorously? 

Reviewer #1: Yes

4. Have the authors made all data underlying the findings in their manuscript fully available?

Reviewer #1: Yes

5. Is the manuscript presented in an intelligible fashion and written in standard English?

Reviewer #1: Yes

6. Review Comments to the Author

Reviewer #1: The authors have addressed most of my comments.

One remaining comment: Please provide info in Abstract and in the Result section re p value (trend) when presenting the decreased likelihood for death in each of the three categories (1. at least one in-person visit. 2. Video only. 3. Telephone only) with ref categories no visits. OR and Conf Intervals are provided for each of the three categories. However, no stat data is provided across the three categories.

It is stated that data will be shared upon reasonable request. I assume that satisfies the Journal's policy.

7. PLOS authors have the option to publish the peer review history of their article (what does this mean? ). If published, this will include your full peer review and any attached files.

**Do you want your identity to be public for this peer review?** For information about this choice, including consent withdrawal, please see our Privacy Policy .

Reviewer #1: **Yes: ** Bengt B. Arnetz

---

## [Author Response · Author response to Decision Letter 3]

26 Feb 2025

February 26, 2025

Rebuttal letter

Dear Members of the Editorial Staff & Reviewers, PLoS One,

We are pleased to re-submit our manuscript entitled, “Outpatient care changes and associated mortality among Veterans with heart failure during the COVID-19 pandemic” for consideration for publication in PLoS One. We have carefully reviewed all comments made by the Editorial Board and Reviewers and revised our manuscript accordingly. Please see below for responses to the comments and suggestions made by the Editorial Board and Reviewers. All suggested changes have been made in the re-submitted manuscript.

Reviewer:

1) Please provide info in Abstract and in the Result section re p value (trend) when presenting the decreased likelihood for death in each of the three categories (1. at least one in-person visit. 2. Video only. 3. Telephone only) with ref categories no visits. OR and Conf Intervals are provided for each of the three categories. However, no stat data is provided across the three categories.

R/ we have now added the p value for linearity trend of the adjusted log hazards for mortality from in-person, video-only, telephone-only to no-visits in both the results section and the abstract.

On behalf of the co-authors, we sincerely thank the reviewer for the thoughtful insight.

Wen-Chih Wu, MD, MPH

---

## [Decision Letter · Decision Letter 3]

6 Apr 2025

Outpatient care changes and associated mortality among Veterans with heart failure during the COVID-19 pandemic

PONE-D-24-06893R3

Dear Dr. Wu,

We’re pleased to inform you that your manuscript has been judged scientifically suitable for publication and will be formally accepted for publication once it meets all outstanding technical requirements.

Kind regards,

Pracheth Raghuveer, MD, DNB

Academic Editor

PLOS ONE

Additional Editor Comments (optional):

Reviewers' comments:

Reviewer's Responses to Questions

**Comments to the Author**

1. If the authors have adequately addressed your comments raised in a previous round of review and you feel that this manuscript is now acceptable for publication, you may indicate that here to bypass the “Comments to the Author” section, enter your conflict of interest statement in the “Confidential to Editor” section, and submit your "Accept" recommendation.

Reviewer #4: All comments have been addressed

Reviewer #5: All comments have been addressed

2. Is the manuscript technically sound, and do the data support the conclusions?

Reviewer #4: Yes

Reviewer #5: Yes

3. Has the statistical analysis been performed appropriately and rigorously? 

Reviewer #4: Yes

Reviewer #5: Yes

4. Have the authors made all data underlying the findings in their manuscript fully available?

Reviewer #4: Yes

Reviewer #5: Yes

5. Is the manuscript presented in an intelligible fashion and written in standard English?

Reviewer #4: Yes

Reviewer #5: Yes

6. Review Comments to the Author

Reviewer #4: it is a well written article . The study has a comprehensive dataset (509,511 patients) providing a detailed comparison of pre-pandemic and pandemic outpatient visits and mortality rates. Inclusion of historical cohorts (2018 & 2019) strengthens the findings by ensuring results are not exclusive to the COVID-19 period. The study design is well defined and statistical approach i s clear and well explained.

However for areas of improvement, I have few comments:

-Many sentences are long and complex, making it harder to follow. Suggest breaking down long paragraphs into smaller, more digestible sections.

-Confounding factors like access to telehealth services (e.g., internet availability, digital literacy), differences in healthcare-seeking behavior among patient subgroups can also be considered.

Overall it is a well done paper and I wish the team , Good luck!

Reviewer #5: Abstract does not have clarity .It may be rewritten for better clarity .It is better to clarify that personal visit once in 6 months reduced mortality hazard by a percentage while video visits reduced the hazard by -- and telephone only visits reduced ---so much

It is better to call video visists as vdeocalls rather than calling it as video and telephone .

grooups could be in person visit, videocall visits and telephone only visits or no visits

7. PLOS authors have the option to publish the peer review history of their article (what does this mean? ). If published, this will include your full peer review and any attached files.

**Do you want your identity to be public for this peer review?** For information about this choice, including consent withdrawal, please see our Privacy Policy .

Reviewer #4: No

Reviewer #5: No

---

## [Editor Report · Acceptance letter]

PONE-D-24-06893R3

PLOS ONE

Dear Dr. Wu,

I'm pleased to inform you that your manuscript has been deemed suitable for publication in PLOS ONE. Congratulations! Your manuscript is now being handed over to our production team.

Kind regards,

on behalf of

Dr. Pracheth Raghuveer

Academic Editor

PLOS ONE